# The Influence of Alumina Bubbles on the Properties of Lightweight Corundum–Spinel Refractory

**DOI:** 10.3390/ma16175908

**Published:** 2023-08-29

**Authors:** Yalou Xin, Yunling Jian, Hongfeng Yin, Yun Tang, Hudie Yuan, Yuchi Liu

**Affiliations:** College of Materials Science & Engineering, Xi’an University of Architecture & Technology, Xi’an 710055, China; xinyalou@xauat.edu.cn (Y.X.); jianyunling@xauat.edu.cn (Y.J.); tangyun@xauat.edu.cn (Y.T.); yuanhudie@xauat.edu.cn (H.Y.); liuyuchi@xauat.edu.cn (Y.L.)

**Keywords:** alumina bubble, lightweight corundum–spinel refractory, compressive strength, refractoriness under load, thermal conductivity

## Abstract

The use of a lightweight corundum–spinel refractory in working lining could reduce the thermal conductivity of industrial furnaces. In this study, bubble alumina was introduced to realize a lightweight Al_2_O_3_-MgAl_2_O_4_ refractory assisted by the reactive sintering of Al_2_O_3_ and MgO. The effects of alumina bubble content and sintering temperature on the phase compositions, microstructure and properties of the lightweight refractory were investigated. The results indicated that the overall performance of the lightweight Al_2_O_3_-MgAl_2_O_4_ refractory was mainly dominated by the content of alumina bubbles. The bulk density, compressive strength and thermal conductivity all decreased when the alumina bubble content increased from 10 to 30 wt%. Meanwhile, the sintering temperature also significantly affected the properties of the obtained refractory. It is worth noting that specimens fired at 1650 °C achieved a high refractoriness under load (RUL) of more than 1700 °C when alumina bubble content was less than 30 wt%, which was comparable to that of the dense Al_2_O_3_-MgAl_2_O_4_ refractory. The thermal conductivity of the obtained samples was remarkably decreased to no more than 2.13 W/(m·K). In order to overcome the trade-off between the light weight of the refractory and overall performance, it is feasible to adjust the content of alumina bubbles and raise the sintering temperature appropriately.

## 1. Introduction

As a basic material, refractories are widely used in high-temperature industrial furnaces. These high-temperature processes are often associated with the massive consumption of resources and energy [1,2]. Due to the negative impact on the environment of high-temperature processes, demand for energy-saving furnaces has become much more urgent. Traditionally, dense refractory accounted for most of the volume of refractories of furnace linings and possessed high thermal conductivity. Although coupled with the application of heat-insulating refractory materials, a lot of heat loss cannot be effectively avoided in high-temperature industrial kilns. Take cement rotary kiln as an example, of which the temperature of the outer wall is usually not less than 200 °C [3,4]. Therefore, it is difficult to reduce heat loss just relying on the improvement of thermal insulation refractory under the current conditions.

Considering the fact that thermal insulating materials closest to the working lining exhibit the best thermal insulating functions [5], the application of a refractory for working lining with a heat insulation function will undoubtedly reduce the heat loss of existing industrial kilns. Driven by this idea, researchers at home and abroad have carried out in-depth and detailed studies and put forward the concept of lightweight refractory materials. This refractory could partially function as insulating material, which effectively achieves energy saving due to its reduced bulk density and thermal conductivity [6,7]. The mechanical properties and erosion resistance can be guaranteed as well. Therefore, it is of great significance to realize a lightweight refractory used in the working lining of industrial furnaces [8,9,10].

Aggregate is an important part of refractory; thus, it is an effective way to realize light weight via preparing microporous aggregate and replacing the traditional dense one with it. And many methods were deployed, such as the decomposition of organic matter, a pore-forming in situ technique and the decomposition of hydrated/carbonated inorganic compounds [11]. Li et al. [12] prepared lightweight Al_2_O_3_-MgO-C refractories through partially replacing traditional tabular alumina aggregates with lightweight alumina aggregates. The results illustrated that the strength of Al_2_O_3_-MgO-C refractories was increased due to optimized interface bonding between the aggregates and the matrix. In addition, the thermal shock resistance and slag corrosion resistance were improved at the same time. Moreover, through using homemade microporous corundum aggregates, lightweight Al_2_O_3_-MgO castables were prepared and exhibited similar or better slag resistance to common Al_2_O_3_-MgO castables [13]. In addition, a vast body of microporous aggregates with different components have been prepared, including MgO-Mg(Al,Fe)_2_O_4_, mullite-corundum, magnesia, magnesium aluminate aggregates, etc. [14,15,16]. It can be obviously seen that research on lightweight refractories mostly focused on the exploitation of lightweight aggregates; however, it is difficult for them to use in castable or refractory bricks on account of the high apparent porosity and large pore size of the prepared lightweight aggregates, finally leading to the deteriorated performance of prepared lightweight refractory products [12,17,18].

To further achieve a trade-off between the light weight and performance of refractories, different methods are also used to achieve the light weight of refractories, avoiding the introduction of microporous aggregates. Taking advantage of the micrometer-scale Kirkendall effect, Liu et al. [19] prepared a lightweight mullite–corundum refractory with high compressive strength (85 MPa), high refractoriness under load (1636 °C) and low thermal conductivity (1.55 W/(m·K) using quartz, bauxite and corundum as the main raw materials. Additionally, they also produced lightweight spinel–corundum refractories through in situ pore formation combined with the Kirkendall effect. And at the same time, the concept of gradient refractories became popular. R. Sarkar et al. [20] prepared a gradient refractory via a simple uniaxial pressing and sintering method. This gradient refractory has a dense part on one side as well as an insulating porous part on the other side to get the two contradictory features in a single refractory brick. In addition, Yin et al. [21] prepared lightweight Al_2_O_3_-MgAl_2_O_4_ refractories through designing a gradient density structure in the matrix due to the inherent competition between the deposition reaction and the mass transport of the gaseous species. In order to further improve the properties of the refractories, carbonate was introduced and achieved higher compressive strength and better thermal shock resistance [22]. But in fact, the aforementioned way to achieve light weight has high requirements for molding and firing, which is not friendly to the existing preparation process of refractory materials. Therefore, it is urgent to develop a facile method to realize a lightweight refractory without the above adverse effects.

It was well-known that alumina bubbles were widely used in thermal insulation materials due to their low thermal conductivity [23,24,25]. On account of its spherical shape and hollow structure, Li et al. [26,27] studied alumina bubble addition on the properties of mullite castables. They found that the flow ability of castables could be improved obviously. Compared to common tabular corundum aggregate, the unique structure and resulting low bulk density and thermal conductivity may also have a positive influence on the light weight of a corundum–spinel refractory. Thus, it is interesting to introduce alumina bubbles as a substitution of tabular corundum aggregate into dense refractories. In the present work, the effect of alumina bubble addition on the bulk density, apparent porosity, compressive strength, refractoriness under load and thermal conductivity of a corundum–spinel refractory has been investigated.

## 2. Experimental

The following raw materials were used for the experiments: activated alumina micro-powder (≥99.5% Al_2_O_3_, Medium diameter: 2.6 μm, Zhengzhou Dengfeng Melting Material Co., Ltd., Zhengzhou, China), fused magnesia powder (≥96.6% MgO, Medium diameter: 13 μm Beijing Lier High-temperature Materials Co., Ltd., Beijing, China), tabular corundum (≥99.2% Al_2_O_3_, 1–3 mm, Almatis Aluminum (Qingdao) Co., Ltd., Qingdao, China) and alumina bubble particle (≥98.5% Al_2_O_3_, 0–1 mm, Kaifeng Gaoda Melting Material Co., Ltd., Kaifeng, China). Figure 1 shows the SEM images of alumina bubble. It can be seen that most of the AB maintained a complete spherical morphology, but there was a certain rate of breakage at the same time. A small quantity of pores existed between grain boundaries. It also can be seen from the fracture surface that the alumina bubble was composed of small alumina crystal. In addition, an ethylene glycol (analytic grade, Tianjin Fuyu Fine Chemical Co., Ltd., Tianjin, China)/phenolic resin (Changzhou Shangkotan Macromolecular Material Co., Ltd., Changzhou, China) solution was used as the binder. The physical properties and chemical compositions of the main raw materials involved in the above process are listed in Table 1 and Table 2, respectively.

In order to prepare lightweight corundum–spinel refractory, first, activated alumina micro-powder and fused magnesia powder were weighted at a molar ratio of 1:1 and then mixed for 4 h in polyurethane pots using alumina balls as grinding media. The obtained mixture was used as the matrix at a mass fraction of 40%, and the other 60 wt% aggregates consisted of alumina bubbles and tabular corundum according to a certain mass ratio. The samples involved in this study are listed in Table 3. Then, the mixtures of matrix and aggregates were pressed into cylinders (Φ36 × 36 mm and Φ36 × 50 mm) under a pressure of 150 MPa. Specifically, samples with the size of Φ36 × 36 mm were used to test the apparent porosity, bulk density and compressive strength according to GB/T2997-2015. Φ36 × 50 mm samples were used to test the refractoriness under load (RUL). Furthermore, cubic samples (240 × 115 × 65 mm) were also prepared for the measurement of thermal conductivity. Eventually, the green compacts were dried at 110 °C for 24 h. Finally, in an electric furnace, the samples were fired at 1550 °C, 1600 °C or 1650 °C for 4 h, respectively.

The phase compositions of samples were analyzed via X-ray diffraction (XRD, D/max2000PC, Nagoya, Japan). The microstructure of fracture surface was observed using a scanning electron microscope (SEM, Hitachi S4800, Tokyo, Japan). The apparent porosity and bulk density of the obtained samples were measured based on the Archimedes principle using water as the medium. A digital pressure tester (YES-600, Jinan, China) was employed to measure the compressive strength. Refractoriness under load (RUL) was tested according to Chinese national standard GB/T5989-2008 [28]. The hot wire method was used to measure thermal conductivity according to ISO 8894-2:1990 [29].

## 3. Results and Discussion

### 3.1. Phase Compositions

The XRD patterns of sample CS10 fired at different temperatures are shown in Figure 2. The result indicated that corundum and spinel were the predominant phases in samples.

Additionally, no other phases were detected. It may be related to the fact that when the sintering temperature was above 1550 °C, the magnesium oxide in the matrix reacted completely with activated alumina micro-powder or other alumina-containing components, such as tabular corundum and alumina bubble, to form spinel. With the increase in temperature, the relative intensity of the spinel diffraction peak increased (marked with the black circles). This effect may be attributed to the formation of more spinel at higher temperature.

Figure 3 shows the XRD patterns of the samples fired at 1650 °C. It was found that the phase composition of the specimens with different contents of alumina bubble did not change significantly, which may suggest that the alumina bubble was not excessively involved in the reaction as Al_2_O_3_-containing ingredients with magnesium oxide to form spinel. This is because usually alumina bubbles are prepared at much higher temperatures of more than 2200 °C. When fired at set temperatures in our experiment, most of them were still preserved in the form of a corundum phase.

### 3.2. Physical Properties

Figure 4 shows the apparent porosity (AP) and bulk density (BD) exhibited by the cylinder samples considered in this work. The results showed that the physical properties were mainly dominated by the content of alumina bubbles. The bulk density of the samples decreased significantly with the increase in alumina bubble content, but the change tendency in apparent porosity was directly reversed. This is because the alumina bubbles possess a hollow structure, which is beneficial for reducing the bulk density of the obtained samples. Meanwhile, the increase in porosity may be related to the breaking of alumina bubbles during the process of molding, because compared with the corundum aggregates, the strength of the alumina bubbles was relatively lower, which caused them to be easily damaged. The breaking of alumina bubbles will undoubtedly lead to an increase in apparent porosity. When the content of alumina bubbles was 10 wt%, specimens had a maximum bulk density of 2.65 g·cm^−3^. However, this value still showed great potential for realizing a lightweight refractory compared to ≥2.85 g·cm^−3^ of conventional dense corundum–spinel refractories.

### 3.3. Compressive Strength

Figure 5 shows the cold compressive strength exhibited by the samples. The influences of sintering temperature and alumina bubble content on compressive strength have been determined. It was found that the compressive strength decreased significantly with the increase in alumina bubble content. This was consistent with the results of the apparent porosity and bulk density mentioned above. Meanwhile, increasing the sintering temperature was helpful for improving the mechanical property. When alumina bubble content was 10 wt%, specimens fired at 1650 °C showed the best compressive strength of 61.4 ± 1.7 MPa. In order to meet the requirements of light weight while ensuring the necessary mechanical properties, the sintering temperature should not be lower than 1600 °C when alumina bubbles are introduced.

Moreover, it is worth noting that there was an obvious downward trend when the content of alumina bubbles varied from 10 wt% to 20 wt%, and with the continuous increase in alumina bubble content, this trend slowed down. For instance, when the change in alumina bubble content was in the first stage, the strength of the sample fired at 1650 °C decreased by 8.8 MPa. The percentage of strength reduction was 14.3%. Correspondingly, the decline in the second stage was 4.1 MPa and 7.8%, respectively. This may be related to the different packing states of alumina bubbles in the samples. When the addition was no more than 20 wt%, as shown in Figure 6a,b, the distribution of alumina bubbles in samples was relatively dispersed. Most of them were kept in a complete sphere shape, which can be regarded as closed pores in the refractory. It could result in a remarkable reduction in compressive strength. However, when the alumina bubble content was higher than 20 wt%, as shown in Figure 6c, the breakage of alumina bubbles was aggravated; the broken alumina bubbles existed in the form of thin flakes in the specimen, which could not cause a significant increase in the closed porosity, so the decline in compressive strength slowed down. It suggested that the appropriate amount of alumina bubbles is not more than 20% to maintain the integrity and avoid breaking.

### 3.4. Performance in Use

In order to evaluate the performance of the obtained lightweight refractory at high temperatures, the refractoriness under load (RUL) was measured. When fired at 1650 °C, the RUL of sample CS30 was 1626.4 °C. In contrast, the RULs of other samples with less AB addition were both higher than 1700 °C, which was comparable to that of dense refractory materials. The high RUL demonstrated that samples with alumina bubble addition showed good resistance to load at high temperature. The reason why the RUL is so high after the introduction of AB may be related to the fact that AB itself has a high RUL.

Thermal conductivity was also measured in order to determine the energy-saving effect of the lightweight refractory. The relationship between thermal conductivity and AB content is illustrated in Table 4. When the alumina bubble content was 10 wt%, the value of thermal conductivity was 2.13 W/(m·K). The value of the dense refractory was nearly 3.2–3.3 W/(m·K). Therefore, the obtained lightweight corundum–spinel refractory possessed great advantages in terms of thermal insulation. It can also be seen that the thermal conductivity decreased significantly with the increase in the alumina bubble content, which was consistent with the increase in apparent porosity. The high porosity increased the gas–solid interface and the phonon propagation of heat conduction in the solid phase, and that led to a decrease in thermal conductivity. In addition, alumina bubbles with a unique hollow structure possessed a much lower thermal conductivity compared to dense-structured tabular corundum aggregates; thus, the existence of alumina bubbles was beneficial for reducing the whole thermal conductivity of the prepared refractories.

### 3.5. Fracture Morphologies

Microstructure evolution was studied using SEM for a better understanding of the variations in the properties mentioned above. Given the hollow structure and resulting low strength compared to tabular corundum aggregate, it is inevitable that the breaking of alumina bubbles was more likely to occur in the molding process. Figure 7 shows the fracture surface of the green compact. Different crushing situations can be clearly seen when the alumina bubble content varied from 10 wt% to 20 wt%. Especially when the amount of alumina bubbles exceeded 20%, due to its high volume fraction and resultant aggregation in the refractory, the crushing of alumina bubbles was further aggravated in the molding process, as shown in Figure 7c. At the same time, although the breakage of alumina bubbles was ubiquitous, some of them were still kept in a relatively complete spherical structure or existed in the morphology of mutual overlap between broken alumina bubble flakes. Both of them were beneficial for realizing the light weight of the refractory. Moreover, the debonding of alumina bubbles in the green body can also be seen in Figure 7d, which indicates that no strong bond existed between the matrix and alumina bubbles.

In contrast, the microstructure of the specimen fired at 1650 °C with 10 wt% alumina bubble addition can be seen in Figure 8. The transgranular fracture of alumina bubbles was observed in fired samples, which indicated that a strong ceramic bond had developed between the alumina bubbles and matrix through sintering. And through observing the internal morphology (red rectangle) of the fractured alumina bubbles, there was an obvious difference compared with the morphology of the matrix (yellow rectangle). It meant that the breaking of the alumina bubbles most likely occurred when the specimen was destroyed, and before the destruction of sample, it was probably preserved as a complete sphere shape. Further observation found that no obvious cracks can be seen between the alumina bubbles and the matrix, showing its good bonding. On the one hand, this good bonding is due to the fact that when the magnesia reacted with alumina to form spinel in the matrix, it was always accompanied by a certain volume expansion which inevitably strengthened the bonding between the alumina bubbles and the matrix. On the other hand, the coefficient of thermal expansion of alumina bubbles and in situ-formed spinel matrices is close, which would not lead to cracks due to thermal expansion mismatch in the sintering and cooling process.

## 4. Conclusions

In summary, the prepared lightweight corundum–spinel refractory possessed the guaranteed mechanical properties, as well as low thermal conductivity, which made the lightweight spinel–corundum refractory partially function as an insulating material, reducing the thermal conductivity of the refractory used in working lining.

(1)After the introduction of alumina bubble particles as a substitute of medium-sized tabular corundum aggregate, the bulk density of the refractory decreased dramatically and realized the light weight of the corundum–spinel refractory. The obtained refractory also possessed high compress strength and refractoriness under load.(2)The overall performance of the lightweight refractory was mainly dominated by the content of alumina bubbles. The sintering temperature also had a significant influence on the properties of the refractory; increasing the temperature was beneficial for improving the mechanical properties.

## Figures and Tables

**Figure 1 materials-16-05908-f001:**
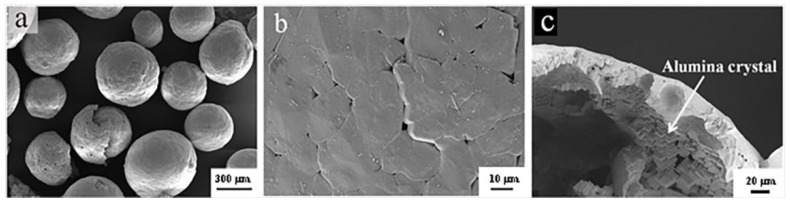
The external morphology (**a**,**b**) and fracture morphology of alumina bubbles (**c**).

**Figure 2 materials-16-05908-f002:**
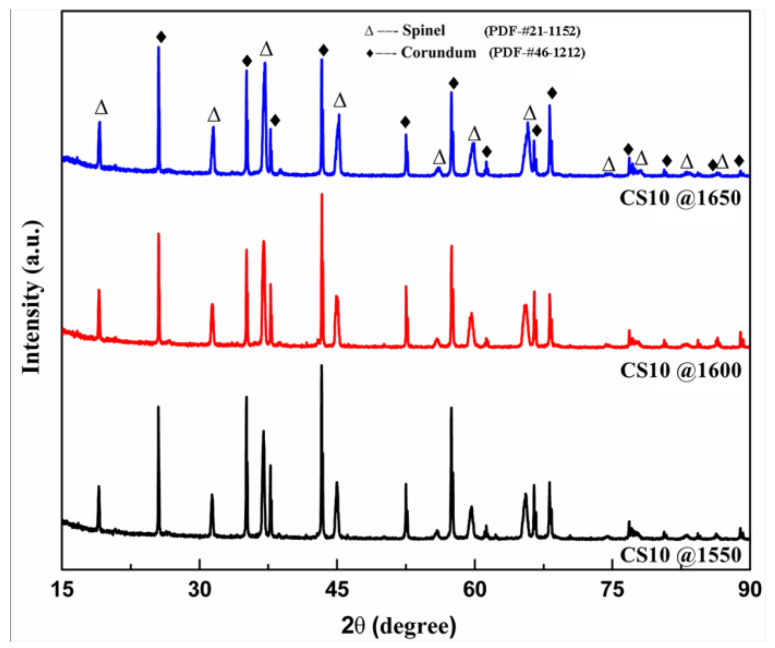
XRD pattern of sample CS10 fired at different temperatures.

**Figure 3 materials-16-05908-f003:**
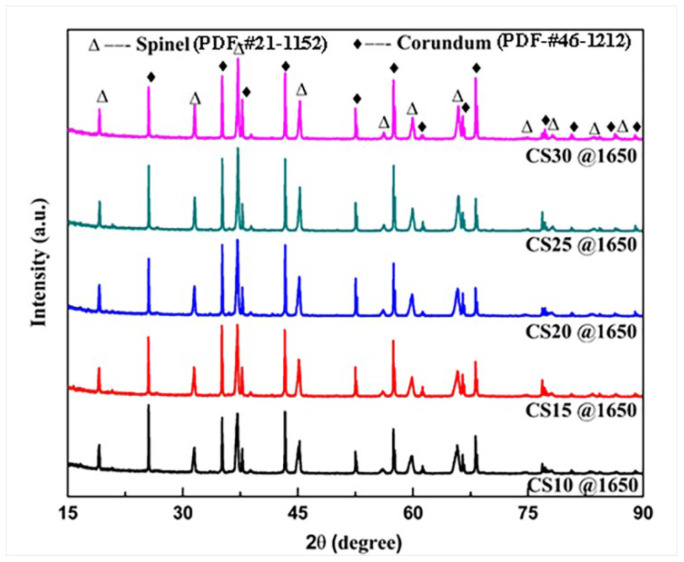
XRD pattern of samples fired at 1650 °C for 4 h.

**Figure 4 materials-16-05908-f004:**
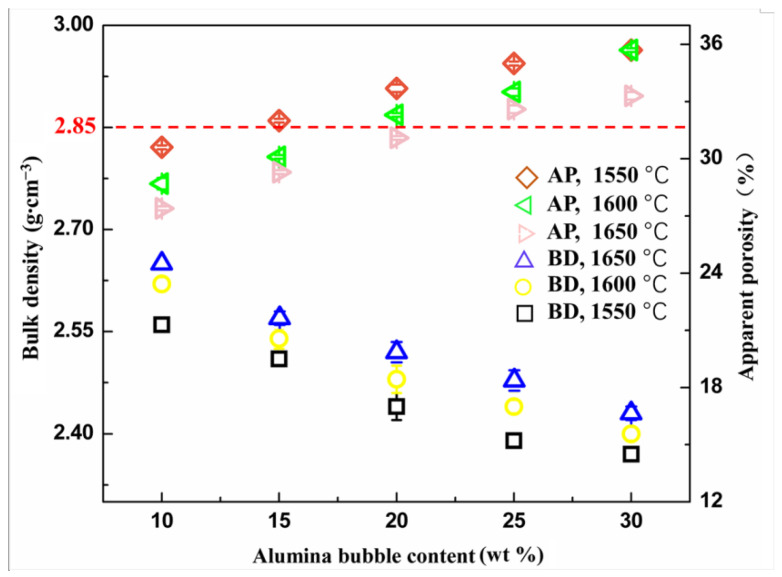
Apparent porosity (AP) and bulk density (BD) for the samples as a function of the alumina bubble content.

**Figure 5 materials-16-05908-f005:**
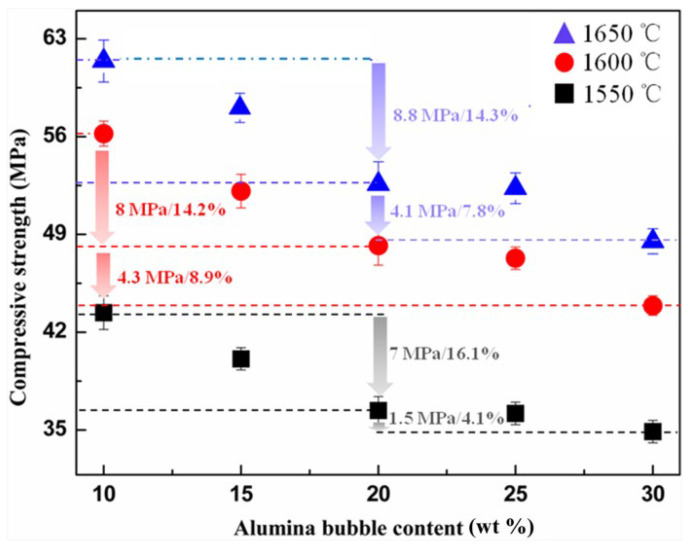
Influences of sintering temperature and alumina bubble content on compressive strength.

**Figure 6 materials-16-05908-f006:**
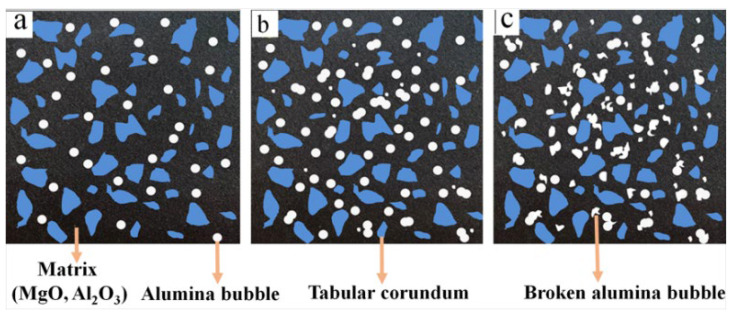
Schematic diagram of alumina bubble distribution in samples: (**a**) 10 wt%, (**b**) 20 wt%, (**c**) 30 wt%.

**Figure 7 materials-16-05908-f007:**
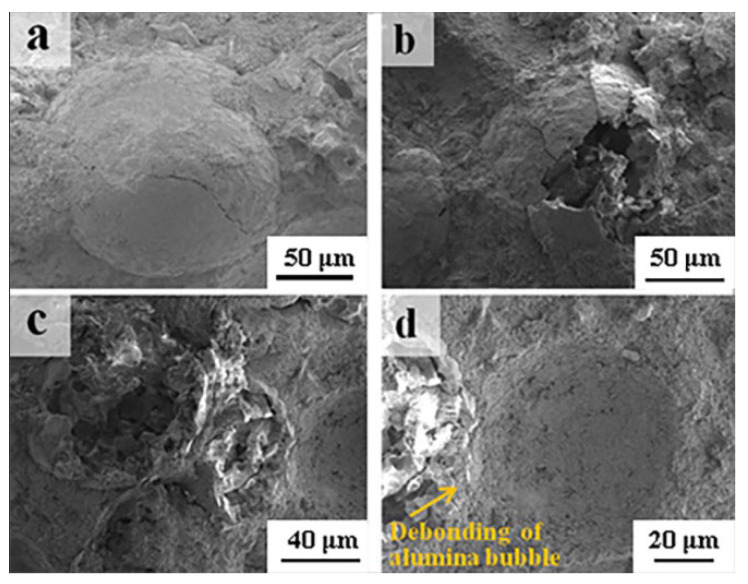
Fracture surface of green compacts with different alumina bubble content: 10 wt% (**a**,**b**), 20 wt% (**c**,**d**).

**Figure 8 materials-16-05908-f008:**
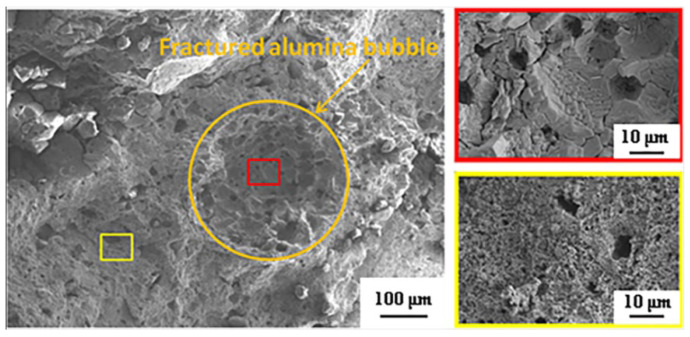
Fracture surface of fired sample sintered at 1650 °C with 10 wt% alumina bubble content.

**Table 1 materials-16-05908-t001:** Physical properties of main raw materials.

	Bulk Density g/cm^3^	Apparent Porosity %	Medium Diameter μm	Breakage Rate %
Activated alumina	≥3.95	—	2.6	—
Fused magnesia	≥3.45	—	13	—
Tabular corundum	≥3.50	≥5	—	—
Alumina bubble	~1.25	—	—	≤15

**Table 2 materials-16-05908-t002:** Chemical composition of main raw materials (wt%).

	Al_2_O_3_	MgO	Na_2_O	CaO	Fe_2_O_3_	SiO_2_	TiO_2_
Activated alumina	≥99.5	≤0.07	≤0.1	≤0.02	≤0.03	≤0.03	—
Fused magnesia	≤0.21	≥96.6	—	≤1.55	≤0.73	≤0.9	≤0.01
Tabular corundum	≥99.2	—	≤0.40	—	≤0.02	≤0.09	—
Alumina bubble	≥98.5	—	—	—	—	—	—

**Table 3 materials-16-05908-t003:** Samples prepared with different preparation parameters.

Sample	Sintering Temperature (°C)	Matrix (wt%)	Aggregate (wt%)	
			Alumina Bubble (0–1 mm)	Tabular Corundum (1–3 mm)
CS10	1550, 1600, 1650	40%	10%	50%
CS15	1550, 1600, 1650	40%	15%	45%
CS20	1550, 1600, 1650	40%	20%	40%
CS25	1550, 1600, 1650	40%	25%	35%
CS30	1550, 1600, 1650	40%	30%	30%

**Table 4 materials-16-05908-t004:** Thermal conductivity of samples sintered at 1650 °C with different AB content.

Sample	Thermal Conductivity (W/(m·K))
CS30@1650	1.52
CS25@1650	1.60
CS20@1650	2.01
CS15@1650	2.10
CS10@1650	2.16

## Data Availability

The data is unavailable due to privacy or ethical restrictions.

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
