# Peer review of "The Influence of Alumina Bubbles on the Properties of Lightweight Corundum–Spinel Refractory"

_materials, 2023, doi:10.3390/ma16175908_

Round 1

Reviewer 1 Report

Title : Effect of Alumina Bubble Addition on the Lightweight and Properties of Corundum-Spinel Refractory.

Dear authors,

The manuscript is good and well written, and the experimental work is appropriate. There are several issues that need to be addressed before the manuscript can be accepted for publication.

Abstract

Line 12 : please replace « firing» by « sintering » thouroughly the manuscript

Lines 21-23, please revise

Introduction

Line 28-29 : please, can authors reformulate the sentense ?

Lines 48-49 : can authors revise the sentensesLine  ?

Lines 56-58 : Moreover,……………. to common Al2O3-MgO castables [11], Please revise to it more readable.

Experimental :

It is mentioned the production of two kind of cylinder samples (Φ36×36 mm and Φ36×50 mm). Can authors provide additional information on the reasons these and their use for the specific performing test with the approriate standard?

Results

XRD :

What is the reference of the PDF card numbers used to identify the potential crystalline phases in the matrix after the various heat treatment cycles?

Lines : 146- 148 : With the increase the sintering temperature……… at higher temperature.  We advise the authors to be careful because the quantification of the phases has not been done. So it could be that this reflection of the peaks are results of the scaling of the combined graphs.

Compressive strength

Cold compressive strength ? what about the dry compressive strenght ? the result described looks more like dry ones.

Lines 183-184 : alumina bubble content was 10 wt%, specimens fired at 1650 °C showed the best compressive strength of 61.4±1.7 MPa. Could authors provide additional information on the obtained findings and discuss their significance?

Fracture morphologies

Lines 237-240 : Especially when…….. , as shown in Figure 7 (c), Could authors rewrite theses sentences ?

Conclusion

Prospects : Are there any specific directions for future research that you would like to suggest based on the results of this study?

Moderate editing of English language required

Author Response

Abstract

Point 1:Line 12 : please replace « firing» by « sintering » thouroughly the manuscript

Response 1: According to reviewer’s comment, we have replace “firing” by “sintering” thouroughly the manuscript. (See Page 1, 5, 6 and 10.)

Point 2:Lines 21-23, please revise                             

Response 2: According to reviewer’s comment, it has been revised. (See Page 1)

Introduction

Point 1:Line 28-29 : please, can authors reformulate the sentense ?

Refractory are widely used in high temperature industries which were usually re- 28 lated to massive consumption of resources and energy.

Response 1: According to reviewer’s comment, it has been revised to “As a basic material, refractories are widely used in high-temperature industrial furnaces. These high temperature processes are often associated with a massive consumption of resources and energy.” (See Page 1)

Point 2:Lines 48-49 : can authors revise the sentensesLine  ?

Response 2: According to reviewer’s comment, it has been revised to “Aggregate is an important part of refractory, thus it is an effective way to realize the lightweight by preparing microporous aggregate and replacing the traditional dense one with it.” (See Page 1)

Point 3:Lines 56-58 : Moreover,……………. to common Al2O3-MgO castables [11], Please revise to it more readable.

Response 3: According to reviewer’s comment, it has been revised to “Aggregate is an important part of refractory, thus it is an effective way to realize the lightweight by preparing microporous aggregate and replacing the traditional dense one with it.” (See Page 2)

Experimental

Point 1:It is mentioned the production of two kind of cylinder samples (Φ36×36 mm and). Can authors provide additional information on the reasons these and their use for the specific performing test with the approriate standard?

Response 1: According to reviewer’s comment, it has been revised.

Samples with the size of Φ36×36 mm are used to test the apparent porosity, bulk density and compressive strength according to GB/T2997-2015. Φ36×50 mm samples are used to test the refractoriness under load (RUL) (See Page 4).

Results

XRD :

Point 1:What is the reference of the PDF card numbers used to identify the potential crystalline phases in the matrix after the various heat treatment cycles?

Response 1: Thanks for the reviewer’s suggestion, it has been revised. (See Figure 2 and Figure 3).

Point 2:Lines : 146- 148 : With the increase the sintering temperature……… at higher temperature.  We advise the authors to be careful because the quantification of the phases has not been done. So it could be that this reflection of the peaks are results of the scaling of the combined graphs.

Response 2: Thanks for the reviewer’s suggestion, it has been revised. (See Page 5).

The quantification of the phases was not calculated, the scaling of the combined graphs was not changed at the same time. The reason why we made this judgment about the increase of spinel content is that we focus more on the relative intensity of the spinel diffraction peak. With the increase of the sintering temperature, the relative intensity of the spinel diffraction peak in each individual diffraction pattern increased as well. So we made such a deduction that “This effect may be attributed to the formation of more spinel at higher temperature”.

Compressive strength

Point 3:Cold compressive strength ? what about the dry compressive strenght ? the result described looks more like dry ones.

Response 3: Thanks for the reviewer’s suggestion. In the field of refractories, because there is not only performance at room temperature, but also high temperature performance, so here we call it cold compressive strength to distinguish it from high temperature mechnical strength, but in fact, it is dry strength.

Point 4:Lines 183-184 : alumina bubble content was 10 wt%, specimens fired at 1650 °C showed the best compressive strength of 61.4±1.7 MPa. Could authors provide additional information on the obtained findings and discuss their significance?

Response 4: Thanks for the reviewer’s suggestion. In this part, we mainly discuss the compressive strength of refractories, and the additional information is not discussed here. This is because in the following paper, we have carried out an in-depth study on the crushing condition and fracture morphologies of alumina balls.

Fracture morphologies

Point 5:Lines 237-240 : Especially when…….. , as shown in Figure 7 (c), Could authors rewrite theses sentences ?

Response 5: According to reviewer’s comment, it has been revised to “Especially when the amount of alumina bubble exceeded 20%, Due to its high volume fraction and resulant aggregation in the refractory, the crushing of alumina bubble was further aggravated in the molding process, as shown in Figure 7 (c)”.

Conclusion

Point 1:Prospects : Are there any specific directions for future research that you would like to suggest based on the results of this study?

Response 1: Thanks for the reviewer’s suggestion. In the following study, we considered to introduce alumina bubbles with a smaller particle size into refractories, which is helpful to avoid a rapid decline in strength even if there is breakage. In addition, alumina bubbles are also considered for use in other refractories, such as corundum-mullite refractories, to achieve the lightweight of refractories.

Reviewer 2 Report

This paper studies the effects of alumina bubbles on the properties of lightweight refractory composites. In my opinion the article is well prepared, and well-structured. There is room for some minor improvements, after which it can be accepted for publication:

1)     The title does not seem to be the correct representation of the work. I mean, its English and sentence structure must be revised.

2)     Introduction lacks the details on the significance of alumina additives. Consider engineering performance and economic significance.

3)     Line 70: 85.0 MPa can be written as 85 MPa.

4)     In table 3: First column, the temperature unit should also be given.

5)     If feasible present the compressive strength results using a bar chart.

6)     Fracture morphology section lacks discussion in the context of available literature.

7)     Conclusion section can be further refined into short and crisp bullet points, stating only the key findings, rather than repetition of what is already discussed above in the results and discussion section.

Minor corrections needed.

Author Response

Point 1: The title does not seem to be the correct representation of the work. I mean, its English and sentence structure must be revised.

Response 1: According to reviewer’s comment, the title has been revised to “The Influence of Alumina Bubble on the Properties of Lightweight Corundum-Spinel Refractory”. (See Page 1)

Point 2: Introduction lacks the details on the significance of alumina additives. Consider engineering performance and economic significance.

Response 2: Thanks for the reviewer’s suggestion. As a basic material, alumina has been widely used as a raw material for refractories. The focus of this article is to discuss the influence of alumina bubbles, used for thermal insulation refractories previously, on the lightweight and properties of corundum-spinel refractory. Thus the significance of alumina additives are not emphasized particularly.

Point 3: Line 70: 85.0 MPa can be written as 85 MPa.

Response 3: According to reviewer’s comment, the title has been revised (See Page 2)

Point 4: In table 3: First column, the temperature unit should also be given.

Response 4: According to reviewer’s comment, it has been revised (See Page 4)

Point 5: If feasible present the compressive strength results using a bar chart.

Response 5: Thanks for the reviewer’s suggestion. We tried to draw a bar chart, but it wasn't pretty, so we did not change this graph. In present graph, we can clearly see the variation trend of compressive strength, as shown by the arrows.

Point 6: Fracture morphology section lacks discussion in the context of available literature.

Response 6: In this paper, the addition of alumina bubbles is high, so the damage of them is serious. This damage is directly related to the change in the performance of refractories. We try to explain the relationship between the damage and the change in performance, and some in-depth analysis and discussion is missing. In the later work (small particle size and low content of alumina bubbles were introduced), we will further supplement and have an in-depth discussion.

Point 7: Conclusion section can be further refined into short and crisp bullet points, stating only the key findings, rather than repetition of what is already discussed above in the results and discussion section.

Response 7: According to reviewer’s comment, the Conclusion section has been further refined into short points (See Pages 9 and 10)

Reviewer 3 Report

The authors show that the thermal conductivity and the apparent porosity of the lightweight corundum-spinel refractory increases by the introduction of alumina bubbles. Moreover, the bulk density and compressive strength decrease with the increase of alumina bubble content.

The authors are kindly asked to do some corrections:

References [12] and [18] are identical, please delete the last one

At line 21, line 71, line 220, line 221, line 282, in Table 4,  etc. please use for Kelvin capital K  so the unit for thermal conductivity is W / (m K)

Author Response

Point 1: References [12] and [18] are identical, please delete the last one

Response 1: Thanks for the reviewer. The error has been corrected.

Point 2: At line 21, line 71, line 220, line 221, line 282, in Table 4, etc. please use for Kelvin capital K, so the unit for thermal conductivity is W / (m K)

Response 2: According to reviewer’s comment, it has been revised.( See Pages 1, 2, 8)